# Broken translational symmetry at edges of high-temperature superconductors

P. Holmvall [1], A.B. Vorontsov [2], M. Fogelström [1] & T. Löfwander [1]

Flat bands of zero-energy states at the edges of quantum materials have a topological origin. However, their presence is energetically unfavorable. If there is a mechanism to shift the band to finite energies, a phase transition can occur. Here we study high-temperature superconductors hosting flat bands of midgap Andreev surface states. In a second-order phase transition at roughly a fifth of the superconducting transition temperature, time-reversal symmetry and continuous translational symmetry along the edge are spontaneously broken. In an external magnetic field, only translational symmetry is broken. We identify the order parameter as the superfluid momentum $\mathbf{p}_s$, that forms a planar vector field with defects, including edge sources and sinks. The critical points of the vector field satisfy a generalized Poincaré-Hopf theorem, relating the sum of Poincaré indices to the Euler characteristic of the system.

---

[1] Department of Microtechnology and Nanoscience-MC2, Chalmers University of Technology, SE-41296 Göteborg, Sweden. [2] Department of Physics, Montana State University, Bozeman, MT 59717, USA. Correspondence and requests for materials should be addressed to T.Löf. (email: tomas.lofwander@chalmers.se)

Superconducting devices are often experimentally realized as thin-film circuits or hybrid structures operating in the mesoscopic regime[1–4]. At this length scale, where the size of the circuit elements becomes comparable with the superconducting coherence length, the nature of the superconducting state may be dictated by various finite-size or surface/interface effects[5]. This holds true in particular for unconventional superconductors, such as the high-temperature superconductors with an order parameter of $d_{x^2-y^2}$ symmetry that changes the sign around the Fermi surface. Scattering at surfaces, or defects, leads to substantial pair breaking and formation of Andreev states with energies within the superconducting gap[6,7]. Today, the material control of high-temperature superconducting films is sufficiently good that many advanced superconducting devices can work at elevated temperatures[8,9]. This raises the question how the specific surface physics of $d$-wave superconductors influences devices.

From a theory point of view, the physics at specular pair-breaking surfaces of $d$-wave superconductors is rich and interesting. The reason is the formation of zero-energy (midgap) Andreev states due to the sign change of the $d$-wave order parameter for quasiparticles scattered at the surface[6,7,10]. In modern terms, there is a flat band of spin-degenerate zero-energy surface states as a function of the parallel component of the momentum, $p_{\parallel}$, which is a good quantum number for a specular surface. A topological invariant has been identified[11,12], that guarantees the flat band for a time-reversal symmetric superconducting order parameter and $p_{\parallel}$ conserved. However, the large spectral weight of these states exactly at zero energy (i.e., at the Fermi energy) is energetically unfavorable. Different scenarios have been proposed, within which there is a low-temperature instability and a phase transition into a time-reversal symmetry-broken phase where the flat band is split to finite energies, thus lowering the free energy of the system. One scenario is the presence of a subdominant pairing interaction and appearance of another order parameter component $\pi/2$ out of phase with the dominant one[13–15], for instance a subdominant $s$-wave resulting in an order parameter combination $\Delta_d + i\Delta_s$. The phase transition is driven by a split of the flat band of Andreev states to $\pm\Delta_s$. The split Andreev states carry current along the surface, which results in a magnetic field that is screened from the bulk. In a second scenario, exchange interactions drive a ferromagnetic transition at the edge where the flat Andreev band is instead spin split[16,17]. A third scenario involves spontaneous appearance of supercurrents[18–20] that Doppler shifts the Andreev states and thereby lowers the free energy. Here the electrons couple to the electromagnetic gauge field $\mathbf{A}(\mathbf{R})$, and this mechanism was first considered theoretically for a translationally invariant edge. In this case, the transition is a result of the interplay of weakly Doppler shifted surface bound states, decaying away from the surface on the scale of the superconducting coherence length $\xi_0$, and weak diamagnetic screening currents, decaying on the scale of the penetration depth $\lambda$. The resulting transition temperature is very low, of order $T^* \sim (\xi_0/\lambda)T_c$, where $T_c$ is the $d$-wave superconducting transition temperature. Later, the transition temperature was shown to be enhanced in a film geometry[21–25] where two parallel pair-breaking edges are separated by a distance of the order of a few coherence lengths. The suppression of the order parameter between the pair-breaking edges can be viewed as an effective Zeeman field that splits the Andreev states and enhances the transition temperature. The mechanism does not involve subdominant channels or coupling to magnetic field, but depends on film thickness $D$, and the transition temperature decays rapidly with increasing thickness as $T^* \sim (\xi_0/D)T_c$.

In this paper, we consider a peculiar scenario[26,27] where spontaneous supercurrents also break translational symmetry along the edge. This scenario too does not rely on any additional

interaction term in the Hamiltonian. Instead, as we will discuss below, it relies on the development of a texture in the gradient of the $d$-wave order parameter phase $\chi$, or more precisely in the gauge invariant superfluid momentum

$$\mathbf{p}_s(\mathbf{R}) = \frac{\hbar}{2}\nabla\chi(\mathbf{R}) - \frac{e}{c}\mathbf{A}(\mathbf{R}), \tag{1}$$

where $\hbar$ is Planck's constant, $e$ the charge of the electron, and $c$ the speed of light. This superfluid momentum spontaneously takes the form of a planar vector field with a chain of sources and sinks along the boundary and saddle points in the interior, see Fig. 1. The vector field is illustrated by arrows showing the local unit vectors $\hat{\mathbf{p}}_s(\mathbf{R})$, while the color scale illustrates the magnitude $p_s(\mathbf{R})$. An interior critical point at $\mathbf{R}_0$ is characterized by a Poincaré index defined as[28,29]

$$I = \frac{1}{2\pi}\oint_\Gamma d\theta, \tag{2}$$

where $\theta = \arctan(p_{sy}/p_{sx})$ is the angle of $\hat{\mathbf{p}}_s(\mathbf{R})$ on the Jordan curve $\Gamma$ encircling $\mathbf{R}_0$. Internal sources and sinks have $I = +1$, while saddle points have $I = -1$. Although the special points on the boundary have to be treated with care, there is a sum rule (Eq. (3)) for the Poincaré indices, as we will discuss below. We identify the $\mathbf{p}_s$ vector field as the order parameter of the symmetry-broken phase, motivated by the fact that the free energy is lowered by a large split of the flat band of Andreev states by a Doppler shift $\mathbf{v}_F \cdot \mathbf{p}_s$, where $\mathbf{v}_F$ is the Fermi velocity. This free energy gain is maximized by maximizing the magnitude of $\mathbf{p}_s$, which is achieved by the peculiar vector field in Fig. 1. The balance of the Doppler shift gain and the energy cost in setting up the vector field with critical points where[30] $\nabla \times \mathbf{p}_s \neq 0$ and the splay patterns between them leads to a high $T^* \approx 0.18T_c$. The inhomogeneous vector field induces a chain of loop-currents at the edge circulating clockwise and anti-clockwise. The induced magnetic fluxes of each loop are a fraction of the flux quantum and form a chain of fluxes with alternating signs along the edge. Here we clarify the structure of the order parameter of the symmetry-broken phase, i.e., $\mathbf{p}_s$, and study the thermodynamics of this phase under the influence of an external magnetic field, explicitly breaking time-reversal symmetry.

## Results

**Translational symmetry breaking in a magnetic field.** In Fig. 2, we show the influence of a rather weak external magnetic field, $B = 0.5B_{g1}$, applied to the $d$-wave superconducting grain with pair-breaking edges for varying temperature near the phase transition temperature $T^*$. The scale $B_{g1} = \Phi_0/\mathcal{A}$ corresponds to one flux quantum threading the grain area $\mathcal{A}$, see the Methods section. The left and right columns show the currents and the magnetic field densities, respectively, induced in response to the applied field. To be concrete, we discuss a few selected sets of model parameters, as listed in Table 1. First, for $T > T^*$ (parameter set I), the expected diamagnetic response of the condensate in the inner part of the grain is present, see Fig. 2a, e. On the other hand, midgap quasiparticle Andreev surface states respond paramagnetically. This situation is well established theoretically and experimentally through measurements of the competition between the diamagnetic and paramagnetic responses seen as a low-temperature up-turn in the penetration depth[18,31,32]. Upon lowering the temperature to $T \gtrsim T^*$ (parameter set II), see Fig. 2b, f, the paramagnetic response at the edge becomes locally suppressed and enhanced, forming a sequence of local minima and maxima in the induced currents and fields. The bulk response is, on the other hand, relatively unaffected. Finally, as $T < T^*$ (parameter set III), see Fig. 2c, g, the regions of minimum current

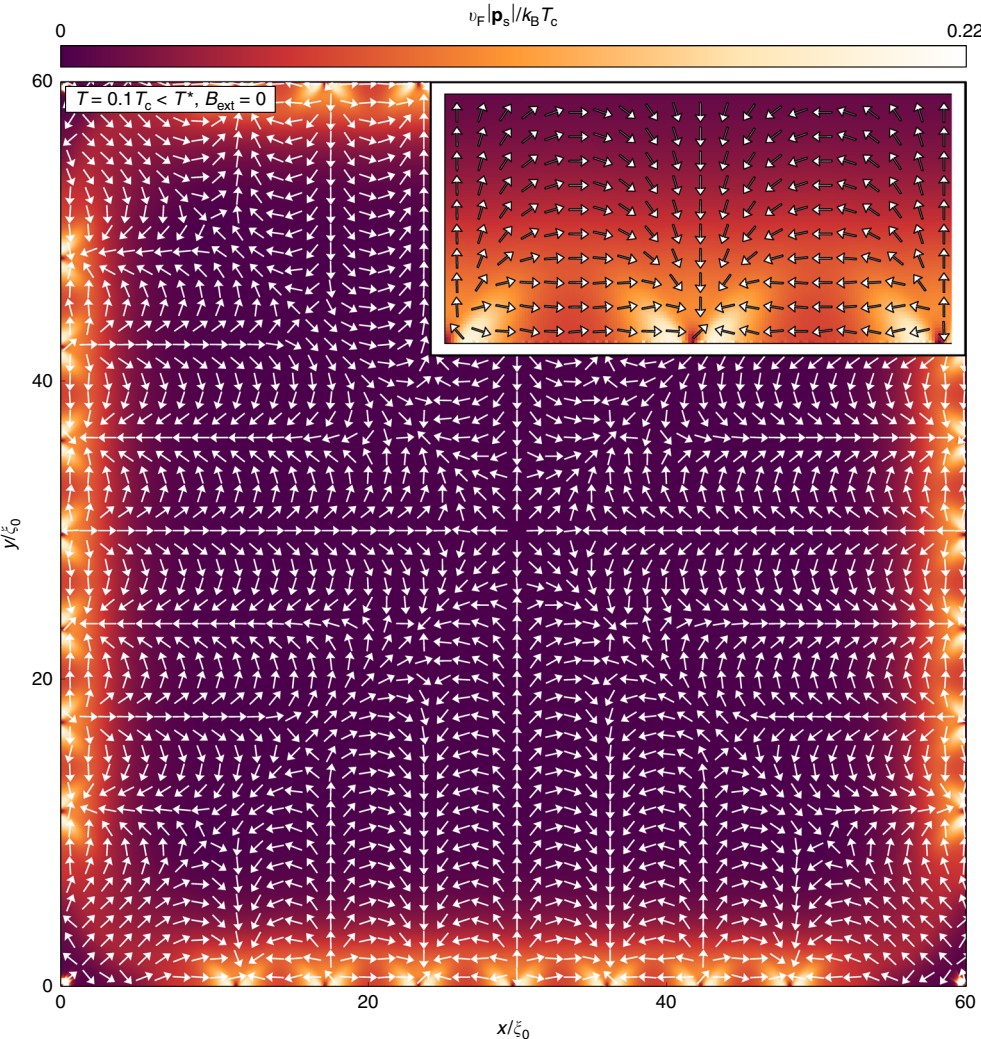

**Fig. 1** Superfluid momentum as a vector field. The superfluid momentum $\mathbf{p}_s$ forms a non-trivial planar vector field with a regular chain of sources and sinks along the edge, thereby breaking local continuous translational symmetry along the edge. Several critical points, including saddle points, sources, and sinks, are formed in the interior. The Poincaré indices of the critical points add up to fulfill the generalized Poincaré-Hopf theorem in Eq. (3). The magnetic field is zero, $B_{ext} = 0$, while the temperature is $T = 0.1T_c$. Since $T$ is well below $T^*$, the splay patterns are rather stiff, leading to triangular shapes near the edges. The stiffness is clear from the magnitude variation shown in color scale. The inset shows one period of the edge structure

turns into regions with reversed currents. The resulting loop currents with clock-wise and anti-clockwise circulations induce magnetic fluxes along the surface with opposite signs between neighboring fluxes. The situation for $T < T^*$ in an external magnetic field can be compared with the one in zero magnetic field[26] displayed in Fig. 2d, h. In the presence of the magnetic field, there is an imbalance between positive and negative fluxes, while in zero external magnetic field, the total induced flux integrated over the grain area is zero.

**Topology of the superfluid momentum vector field**. Let us quantify the symmetry-broken phase in a magnetic field by plotting the superfluid momentum defined in Eq. (1), see Fig. 3. For $T \gtrsim T^*$ (parameter set II), the amplitude of $\mathbf{p}_s$ varies along the edge (coordinate $x$), see Fig. 3a, reflecting the varying paramagnetic response in Fig. 2b, f. For $T < T^*$ (parameter set III), the sources and sinks have appeared pairwise together with a saddle point, see Fig. 3b. The left defects in the figure are not well developed because of the proximity to the corner. Finally, in Fig. 3c, we show the vector field at a lower temperature when the chain of sources, sinks, and saddle points are well established and

the magnitude of $\mathbf{p}_s$ is large, much larger than in the interior part of the grain still experiencing diamagnetism. In a magnetic field, the vector field far from the surface has a preferred direction reflecting the diamagnetic response of the interior grain. This shifts the sources and sinks along the surface, as compared with the regular chain for zero field in Fig. 1, and moves the saddle points to the surface region.

The superflow pattern of sources, sinks, and saddle points satisfy a certain sum rule related to the topology of the sample. This relation also ties the special points of the $\mathbf{p}_s$ field on the edge of the sample with critical points in its bulk. The generalized Poincaré-Hopf theorem for manifolds with boundaries[33,34] connects the properties of a vector field $v$ inside a manifold $M$, and on its boundary $\partial M$, with the Euler characteristic of the manifold $\chi(M)$. Using the formulation presented in ref. [34], we write

$$\mathrm{Ind}_M(v) + \frac{1}{2}\left[\mathrm{Ind}_{\partial_- M}(v_{||}) - \mathrm{Ind}_{\partial_+ M}(v_{||})\right] = \chi(M), \qquad (3)$$

where $\mathrm{Ind}_M(v)$ is the total Poincaré index of critical points of the field $v$ internal to $M$, $\mathrm{Ind}_{\partial_\pm M}(v_{||})$ is the total Poincaré index of

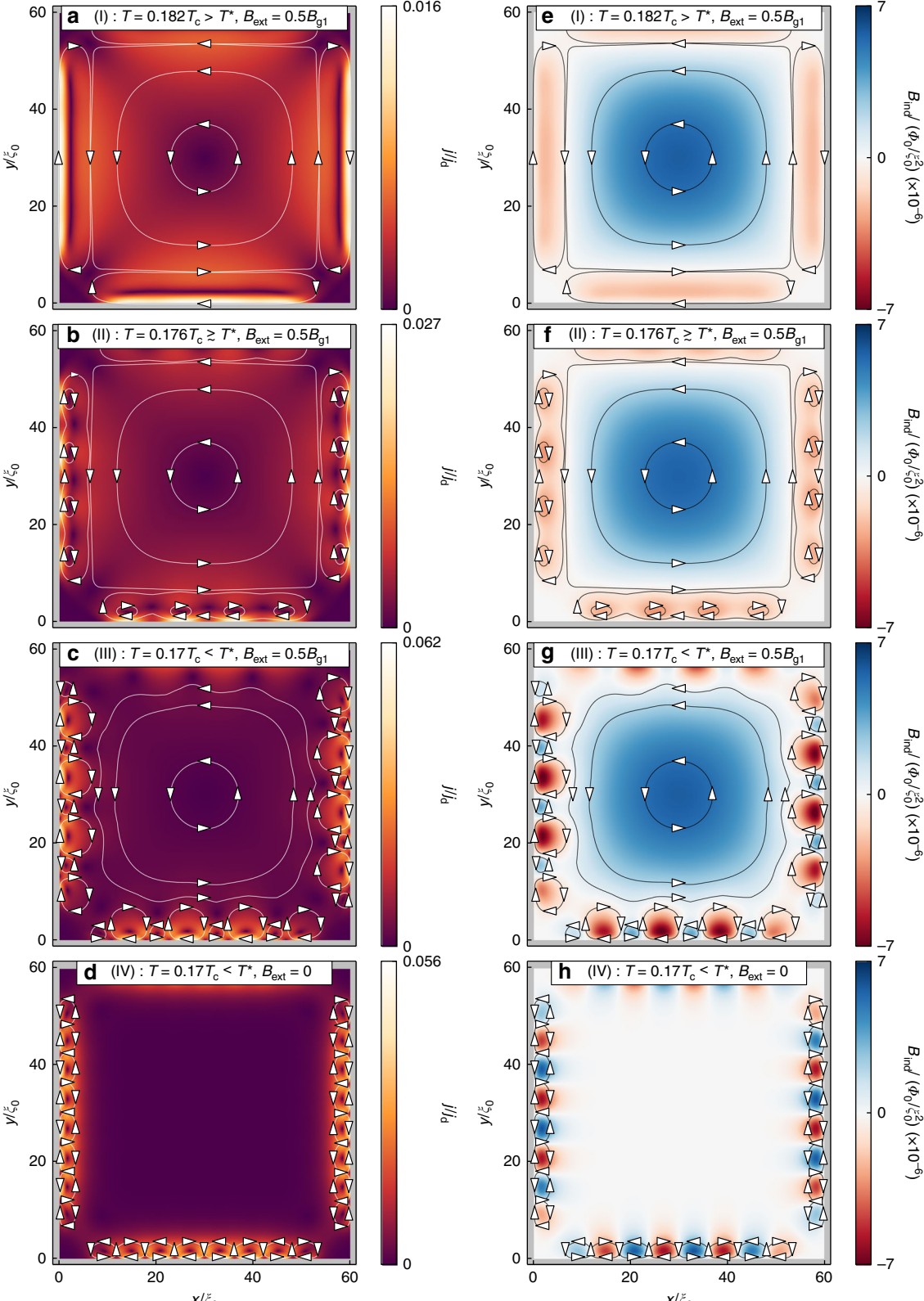

**Fig. 2** Spontaneously formed currents and induced magnetic field. **a–d** Total current magnitude and **e–h** induced magnetic flux density for different temperatures and external fields (see annotations). Lines and arrows have been added to illustrate the flow of the currents

critical points of the tangent vector $v_{||} || \partial M$ on the boundary. The theorem applies when the boundary $\partial M$ does not go through any critical points of $v$. Boundary indices where field points inside

$(\partial_- M)$ / outside $(\partial_+ M)$ of $M$, come with positive/negative signs. In Supplementary Note 1, we demonstrate in detail how the sum rule works for the vector field in Fig. 1, redrawn as a streamline

plot in Supplementary Fig. 2. We also provide other examples of grain geometries in Supplementary Figs. 3–10. We utilize the sum rule as a tool to verify that the calculations are correct.

In a magnetic field, as in Fig. 3, a motif with one edge source, one edge sink, and one saddle point annihilate at $T^*$. In the same fashion, increasing the magnetic field strength, the motif gets smaller as the defects are forced toward each other to match the superflow in the bulk. However, the magnitude of $\mathbf{p}_s$ near the surface due to Meissner screening of the bulk is not large enough to force an annihilation of the motifs. The broken symmetry phase therefore survives the application of an external magnetic field within the whole Meissner state, $b \in [0, 1]$.

For higher fields, when Abrikosov vortices start to enter the grain, the problem quickly becomes complicated by the interplay of the Abrikosov vortex lattice formation and finite grain size effects. The free energy landscape is very flat and it is possible to find multiple metastable configurations. For a variety of grain sizes and magnetic field strengths, we have established coexistence of Abrikosov vortices and the spontaneously formed edge

loop currents[35]. We therefore conclude that the edge loop-current phase established for $T < T^*$ should survive into the mixed state, but a complete investigation of the geometry-dependent phase diagram for large fields is beyond the scope of this paper.

**Induced currents and magnetic fields.** Let us investigate further how the currents and magnetic fields are induced at $T^*$. As we have seen, the paramagnetic response and the spontaneously appearing edge loop currents compete, as they both lead to shifts of midgap Andreev states. As the temperature is lowered, the strength of the paramagnetic response increases slowly and linearly, while the strength of the loop currents increases highly non-linearly. This is illustrated in Fig. 4, by plotting the area-averaged current magnitude

$$\bar{j} = \frac{1}{\mathcal{A}} \int d^2 R |\mathbf{j}(\mathbf{R})|, \tag{4}$$

as a function of temperature for the cases when $B_{ext} = 0$ (solid line), $B_{ext} = 0.5B_{g1}$ (dashed line), and for comparison also for a system without pair-breaking edges having only a diamagnetic response at $B_{ext} = 0.5B_{g1}$ (dash-dotted line). The paramagnetic response is fully suppressed at low temperatures $T < T^*$. Such a sudden disappearance of the paramagnetic response at a temperature $T^*$ should be experimentally measurable, for example in the penetration depth or by using nano-squids[36,37].

We show in Fig. 5a the total induced magnetic flux through the grain

$$\Phi_{ind} = \int d^2 R B_{ind}(\mathbf{R}), \tag{5}$$

**Table 1 Sets of parameters used for presenting results**

| Set | Temperature | External magnetic field |
|---|---|---|
| (I) | $T = 0.182T_c > T^*$ | $B_{ext} = 0.5B_{g1}$ |
| (II) | $T = 0.176T_c \gtrsim T^*$ | $B_{ext} = 0.5B_{g1}$ |
| (III) | $T = 0.17T_c < T^*$ | $B_{ext} = 0.5B_{g1}$ |
| (IV) | $T = 0.17T_c < T^*$ | $B_{ext} = 0$ |

The field scale $B_{g1} = \Phi_0/\mathcal{A}$ corresponds to an external magnetic flux through the grain area exactly equal to one flux quantum

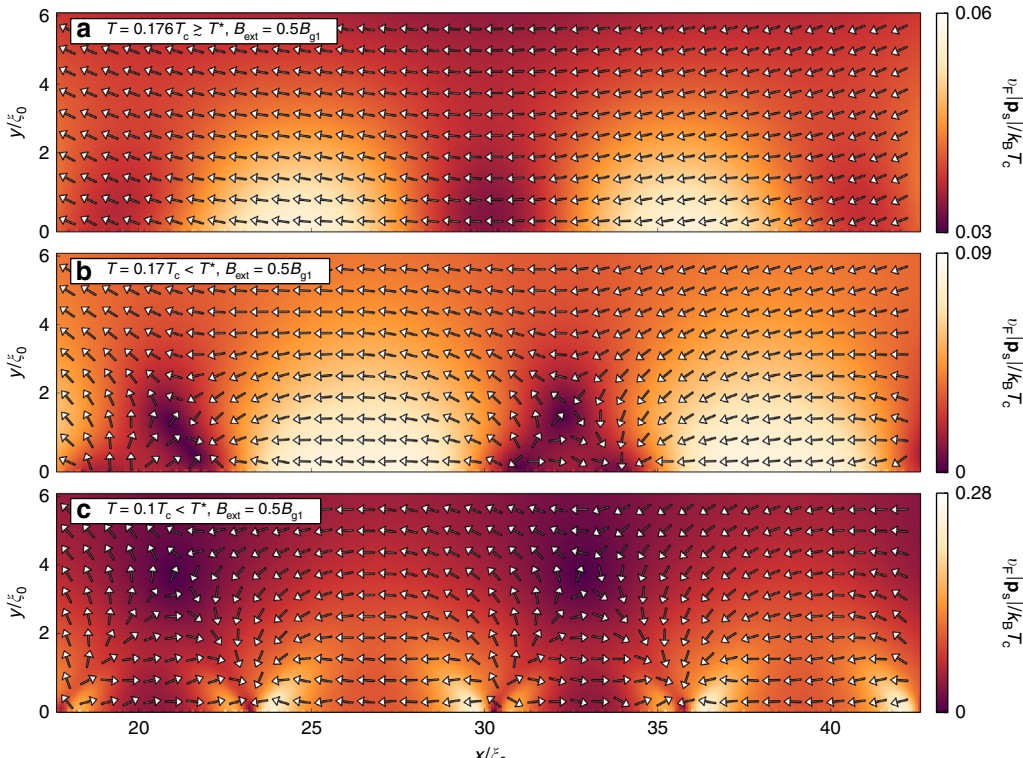

**Fig. 3** Superfluid momentum for varying temperature. **a** The superfluid momentum induced in an external magnetic field of $B_{ext} = 0.5B_{g1}$ for a temperature slightly above the transition temperature $T^*$ reflects the paramagnetic response. **b** At the phase transition, source–sink–saddle-point motifs appear and separate along the edge breaking translational invariance along the edge coordinate $x$. **c** For lower temperature, the magnitude $|\mathbf{p}_s|$ grows large. Note that different color scales are used in the subfigures in order to enhance visibility

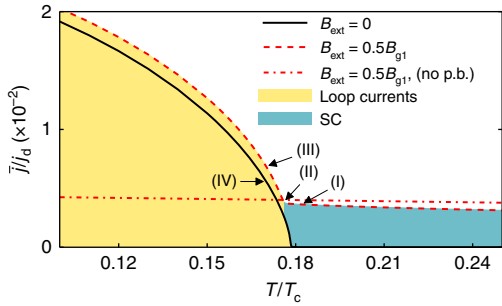

**Fig. 4** Current as a function of temperature. The area-averaged current magnitude, defined in Eq. (4), is plotted for zero external magnetic field (solid line), with an external magnetic field of magnitude $B_{ext} = 0.5B_{g1}$ (dashed line), and for a system without pair-breaking edges at $B_{ext} = 0.5B_{g1}$ (dash-dotted line). In the latter case, the system only displays a diamagnetic response. Letters (I)–(IV) indicate the parameter values corresponding to the fields in Fig. 2, see Table 1

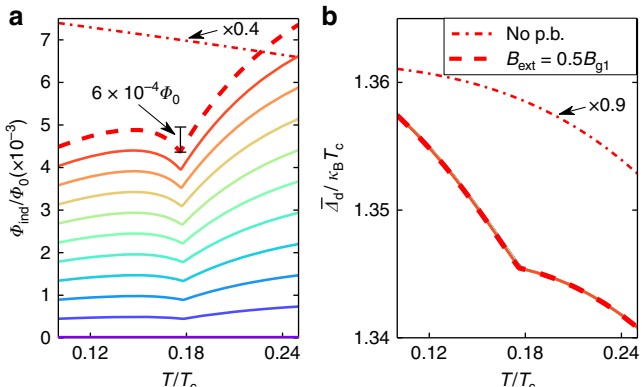

**Fig. 5** Magnetic flux as a function of temperature. **a** Temperature dependence of the induced magnetic flux, defined in Eq. (5). The solid lines indicate, from bottom to top (colors purple to red), the external field magnitude from $B_{ext} = 0$ to $B_{ext} = 0.5B_{g1}$ in steps of $0.05B_{g1}$. The line corresponding to zero field lies exactly at zero since there is an equal amount of positive and negative fluxes induced in this case, see Fig. 2. Panel **b** shows the area-averaged order parameter magnitude defined in Eq. (6) versus temperature. Results are also shown for a system without pair-breaking edges (dash-dotted line) at $B_{ext} = 0.5B_{g1}$, but scaled with a factor 0.4 and 0.9 in **a** and **b**, respectively

and in Fig. 5b the area-averaged order parameter magnitude

$$\overline{\Delta}_d = \frac{1}{\mathcal{A}} \int d^2 R |\Delta_d(\mathbf{R})|, \qquad (6)$$

both as functions of temperature for different values of $B_{ext}$. The figures also show results for a $d$-wave grain without pair-breaking edges at $B_{ext} = 0.5B_{g1}$ (dash-dotted line). For better visibility, the latter results have been scaled by a factor 0.4 and 0.9 in (a) and (b), respectively. Two different trends are distinguishable in the observables for $T < T^*$ and $T > T^*$, separated by a "kink". The induced magnetic flux through the grain area decreases as $T$ decreases down to $T^*$ due to the increasing paramagnetic response that competes with the diamagnetic one. At $T^*$, the inhomogeneous edge state appear and starts competing with the paramagnetic response. Thus, the total magnetic flux increases again. At the same time, the order parameter is partially healed.

**Phase transition and thermodynamics**. The sudden changes with a discontinuity in the derivative as a function of temperature of the total induced current, the magnetic flux, as well as the order parameter (Figs. 4 and 5) indicate that there is a phase transition occurring at the temperature $T^*$. In zero external magnetic field, there is a second-order phase transition at $T^*$, where both time-reversal symmetry and continuous translational symmetry along the edge are spontaneously broken[26]. Let us now investigate the thermodynamics in an external magnetic field already explicitly breaking time-reversal symmetry.

In Fig. 6a, we plot the free energy difference between the superconducting and normal states $\Omega_S - \Omega_N$, defined in Eq. (29), for external field $B = 0.5B_{g1}$ (red dashed line) and for zero field (solid black line). For comparison, we show the free energy difference for a purely real order parameter in zero field (gray fine line), i.e., without the symmetry breaking edge loop currents. For $T < T^*$, this solution is not the global minimum of the free energy, and we therefore refer to it as a metastable state. To enhance the visibility of the differences in free energy between the possible solutions, we show in Fig. 6b the free energy difference with respect to the metastable state, i.e., $\Omega_S - \Omega_{ms}$. The small slope in the red dashed line at $T > T^*$ in Fig. 6b is caused by the shift of midgap Andreev states due to the paramagnetic response, which increases as $T$ decreases. The phase transition temperature $T^*$ for the second-order phase transition can be identified with the "knee" in the entropy difference defined in Eq. (31), see Fig. 6c, d. Since time-reversal symmetry is already explicitly broken by the external magnetic field, the phase transition signals breaking of local continuous translational symmetry and establishment of the vector field $\mathbf{p}_s$ with the chain of defects along the edge, as shown in Fig. 3. The magnitude of the order parameter follows the expected scaling law for second-order phase transitions, $p_s(T) \propto (1 - T/T^*)^{\beta}$ with $\beta = 1/2$, as shown in the inset of Fig. 6d. However, the temperature range within which the scaling law holds is very limited and non-linear terms play an important role for lower temperatures $T < T^*$.

The knee in the entropy leads to a jump in the specific heat, as shown in Fig. 6e, f. The heat capacity is expressed in units of the heat capacity jump at the normal-superconducting phase transition at $T_c$ for a bulk $d$-wave system

$$\Delta C_d = \frac{2\alpha}{3} \mathcal{A} k_B^2 T_c N_F, \qquad (7)$$

where $\alpha = 8\pi^2/[7\zeta(3)]$, with $\zeta$ being the Riemann-zeta function. The jump in heat capacity at the phase transition is an edge-to-area effect, and grows linearly as the sample becomes smaller. The jump is roughly 4.5% of $\Delta C_d$ for the mesoscopic $\mathcal{A} = 60 \times 60\xi_0^2$ grain considered here, and grows as the size of the grain is reduced. The phase transition temperature $T^*$ is extracted as a function of $B_{ext}$ as the midpoint temperature of the jump in the specific heat. Figure 7 shows a phase diagram where the $T^*$, extracted in this way from the specific heat, is plotted versus external field strength (crosses). We compare this with $T^*$ extracted as the minimum (the "kink", see Fig. 5a) in the induced flux. The small lowering of $T^*$ with increased $B_{ext}$ is caused by the competing paramagnetic response.

From the above, it is clear that the phase with edge loop currents shows extreme robustness against an external magnetic field in the whole Meissner region ($B_{ext} \leq B_{g1}$). The magnitude of the spontaneously formed superfluid momentum $\mathbf{p}_s$ at the edge grows non-linearly to be very large for $T < T^*$, fueled by the lowering of the free energy by Doppler shifts of the flat band of Andreev surface states. The corresponding correction to $\mathbf{p}_s$, due to the process of screening of the external magnetic field, is in comparison small. Thereby, $T^*$ is not dramatically shifted in a

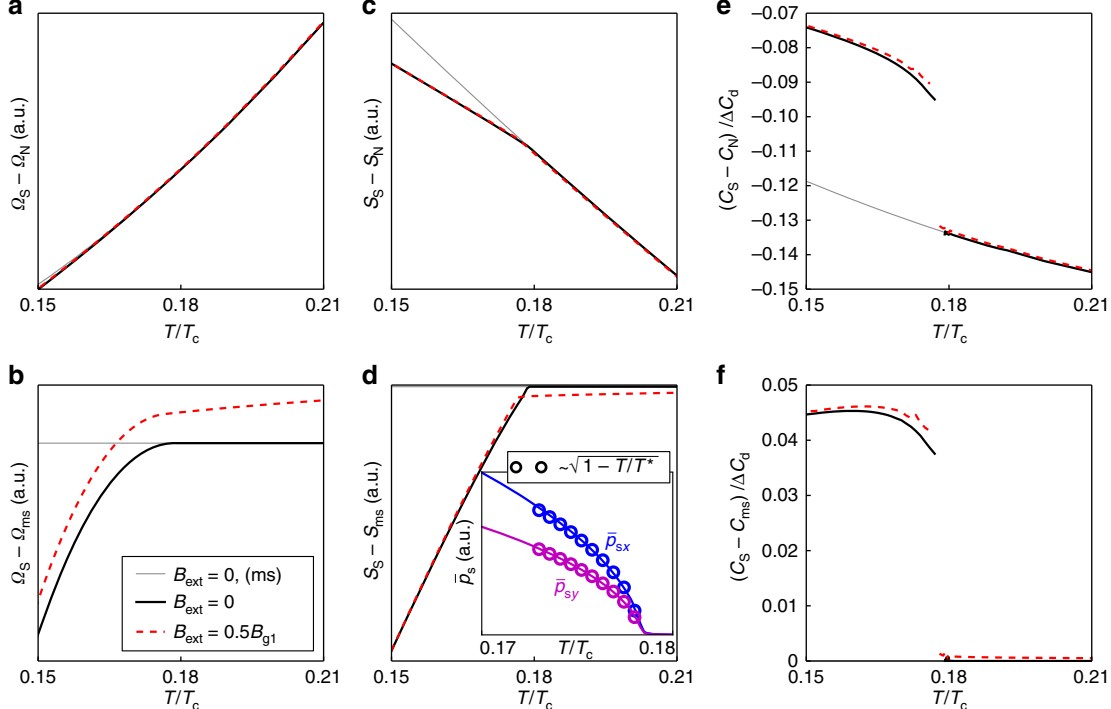

**Fig. 6** Thermodynamics and phase transition. **a**, **b** free energy, **c**, **d** entropy, and **e**, **f** specific heat capacity, versus temperature. The lines correspond to a system with purely real order parameter without edge currents (gray fine line), a system with spontaneous edge currents in zero magnetic field (black solid line), and in a finite external field $B = 0.5B_{g1}$ (red dashed line). In the lower panels **b**, **d**, and **f**, the quantities have been subtracted by the corresponding values of the system with a purely real order parameter, the metastable (ms) state. The heat capacity is normalized by the heat capacity jump at the normal-superconducting phase transition for a bulk $d$-wave system, $\Delta C_d$ in Eq. (7). The inset in (**d**) shows the temperature dependence of the superfluid momentum near $T^*$, averaged over a few source–sink unit cells at one edge. It follows the expected temperature dependence for the order parameter at a mean-field second-order phase transition

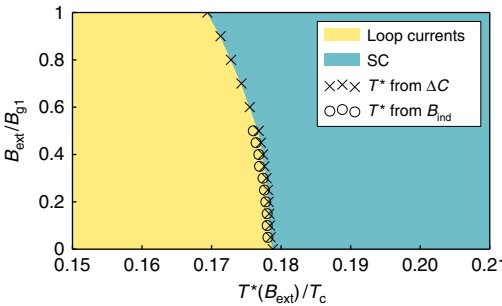

**Fig. 7** Phase diagram. The transition temperature $T^*$ to a state with spontaneously broken continuous translational symmetry is plotted as a function of the external magnetic flux density. The crosses show $T^*$ extracted from the jump in the specific heat in Fig. 6e, while the open circles show $T^*$ extracted from the minimum of the total induced magnetic flux in Fig. 5a

magnetic field and the symmetry-broken phase below $T^*$ is robust.

## Discussion

Which of the scenarios outlined in the introduction wins will ultimately depend on the material properties of a specific high-temperature superconducting sample, or the material properties of other candidate $d$-wave superconductors, e.g., FeSe[38]. In the scenario studied here, the resulting transition temperature is high, $T^* \sim 0.18T_c$. It means that the interaction terms in the

Hamiltonian for the other scenarios would have to be sufficiently large in order to compete. It is even possible that one or another scenario wins in different parts of the material's phase diagram[16].

We note that the phase transition at $T^*$ means that the initially topologically protected flat band of zero energy surface states is shifted away from the Fermi energy. Such fragility of topologically protected states has been studied recently e.g., for topological insulators[39] supporting the quantum spin-Hall state. In that case, an edge reconstruction due to Coulomb interactions leads to breaking of time-reversal symmetry. In the $d$-wave super-conductor case, although the bulk Hamiltonian still maintains required symmetries, a local instability at the surface violates these symmetries spontaneously and moves the flat band of bound states to finite energies. The spontaneously broken trans-lational symmetry allows for a larger shift from zero energy and a high $T^*$.

From an experimental point of view, the surface physics of $d$-wave superconductors is complicated by, for instance, surface roughness, inhomogeneous stoichiometry, and presence of impurities. The formation of a band of Andreev states centered at zero energy is well established by numerous tunneling experi-ments, in agreement with the expectation for $d$-wave symmetry of the order parameter, as reviewed in refs. [6,7]. One consistent experimental result is that the band is typically quite broad, with a width that saturates at low temperature. On the other hand, the establishment of a time-reversal symmetry breaking phase remains under discussion, see for instance refs. [40,41]. Several tunneling experiments on YBCO[42–44] show a split of the zero-bias conductance peak, while others do not[45,46]. Other probes indicating time-reversal symmetry breaking include thermal conductivity[47], Coulomb blockade in nanoscale islands[5], and

STM tunneling at grain boundaries in FeSe[38]. As we argued in refs. [26,27] within the scenario with spontaneous loop currents, the split of the Andreev band might be difficult to resolve in a tunneling experiment because of the broken translational symmetry along the edge and associated variations in the superflow field. This leads to a smearing effect for tunnel contacts with an area larger than the coherence length and an expected wide, largely temperature-independent, peak centered at zero energy. In fact, this would be consistent with most tunneling experiments.

With an eye to inspire a new generation of experiments, we have presented results for the interplay between an external magnetic field, that induces screening supercurrents, and the phase transition at $T^*$ into a state with the spontaneous loop currents at the edges. We have shown that the phase should be quantified in terms of its order parameter, the vector field $\mathbf{p}_s(\mathbf{R})$, which contains edge sources and sinks, as well as saddle points. At all these critical points, $\nabla \times \mathbf{p}_s \neq 0$. The $\mathbf{p}_s$ vector field drives the loop currents with opposite circulations in neighboring loops. The loop-current strength increases highly non-linearly, suppressing the paramagnetic response present for $T > T^*$. As the strength of the external magnetic field increases, the size of the Doppler shift due to the paramagnetic response grows linearly. Therefore, $T^*$ decreases slightly as the magnitude of the external field increases. The influence of the external field, and in particular the sudden disappearance of the paramagnetic response, leads to observables which we argue should be visible in experiment. For example the "kink" in the total induced flux at $T^*$. The magnetic fluxes induced by the loop currents should be directly observable with recently developed scanning probes[36,37], and the sudden disappearance of the paramagnetic response should be observable with nano-SQUIDS and possibly in penetration-depth experiments. Furthermore, the large jump in heat capacity at the phase transition should be observable with nanocalorimetry[48].

The identification of the order parameter $\mathbf{p}_s(\mathbf{R})$, with its topological textures, leads to similarities with other systems, including general relativity[33], fluid dynamics[49], liquid crystals[50], and superfluid $^3$He[51]. An interesting difference is that in those systems, there is typically a transition in a preexisting vector field to a state with topological textures. Here, instead, we have a singlet $d$-wave superconductor that spontaneously establishes $\mathbf{p}_s(\mathbf{R})$ with topological textures different than the traditional Abrikosov vortices.

## Methods

**Model and grain geometry.** Our aim is to investigate the ground state of clean mesoscopic $d$-wave superconducting grains in an external magnetic field applied perpendicular to the crystal $ab$-plane, as shown in Fig. 8. As a typical geometry, we consider a square grain with side lengths $D = 60\xi_0$, where $\xi_0 = \hbar v_F/(2\pi k_B T_c)$ is the zero-temperature superconducting coherence length, $v_F$ is the normal state Fermi velocity, and $k_B$ the Boltzmann constant. The sides of the system are assumed to be misaligned by a 45° rotation with respect to the crystal $ab$-axes, inducing maximal pair-breaking at the edges.

The external field is directed perpendicular to the $xy$-plane,

$$\mathbf{B}_{ext} = -B_{ext}\hat{\mathbf{z}}||\hat{\mathbf{c}}. \tag{8}$$

We shall consider rather small external fields, and will use a field scale $B_{g1} = \Phi_0/\mathcal{A}$, corresponding to one flux quantum threading the grain of area $\mathcal{A} = D^2 = 60\xi_0 \times 60\xi_0$. The flux quantum $\Phi_0 = hc/(2|e|)$ in Gaussian CGS units. The field $B_{g1}$ is larger than the lower critical field $B_{c1} \propto \Phi_0/\lambda_0^2$, where vortices can enter a macroscopically large superconductor, since the grain side length is smaller than the penetration depth. We assume that $\lambda_0 = 100\xi_0$, relevant for YBCO. The upper critical field $B_{c2} \propto \Phi_0/\xi_0^2$ is much larger than any field we include in this study. To be precise, we parameterize the field strength as

$$B_{ext} = bB_{g1}, B_{g1} \equiv \frac{\Phi_0}{\mathcal{A}}, \tag{9}$$

and we will consider $b \in [0, 1]$.

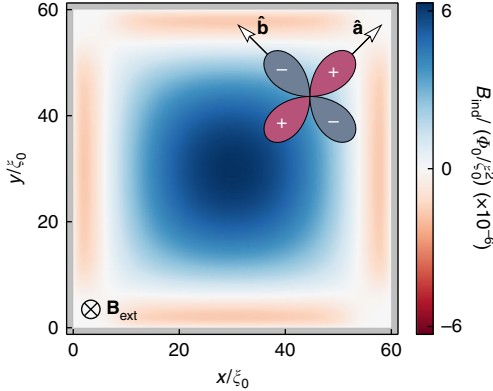

**Fig. 8** Grain geometry. The system consists of a $d$-wave superconducting grain exposed to an external magnetic field $\mathbf{B}_{ext} = -B_{ext}\hat{\mathbf{z}}$. The crystal $ab$-axes are rotated 45° relative to the grain edges, inducing pair breaking at the edges of the system. The color scale shows the magnetic field $B_{ind}$ induced in response to an external field of size $B_{ext} = \Phi_0/2\mathcal{A}$ at a temperature $T = 0.2T_c$. There is a diamagnetic response carried by the condensate in the interior, and a paramagnetic response carried by midgap surface Andreev states at the edges

**Quasiclassical theory.** We utilize the quasiclassical theory of superconductivity[52–54], which is a theory based on a separation of scales[55–58]. For instance, the atomic scale is assumed small compared with the superconducting coherence length, $\hbar/p_F \ll \xi_0$. This separation of scales makes it possible to systematically expand all quantities in small parameters such as $\hbar/p_F\xi_0$, $\Delta/\epsilon_F$, and $k_B T_c/\epsilon_F$, where $\Delta$ is the superconducting order parameter, $p_F$ is the Fermi momentum, and $\epsilon_F$ is the Fermi energy. In equilibrium, the central object of the theory is the quasiclassical Green's function $\hat{g}(\mathbf{p}_F, \mathbf{R}; z)$, which is a function of quasiparticle momentum on the Fermi surface $\mathbf{p}_F$, the quasiparticle center-of-mass coordinate $\mathbf{R}$, and the quasiparticle energy $z$. The latter is real $z = \epsilon + i0^+$ with an infinitesimal imaginary part $i0^+$ for the retarded Green's function, or an imaginary Matsubara energy $z = i\epsilon_n = i\pi k_B T(2n + 1)$ in the Matsubara technique ($n$ is an integer). To keep the notation compact, the dependence on the parameters $\mathbf{p}_F$, $\mathbf{R}$, and $z$ will often not be written out. The hat on $\hat{g}$ denotes Nambu (electron-hole) space

$$\hat{g} = \begin{pmatrix} g & f \\ -\tilde{f} & \tilde{g} \end{pmatrix}, \tag{10}$$

where $g$ and $f$ are the quasiparticle and pair propagators, respectively. The tilde operation denotes particle-hole conjugation

$$\tilde{\alpha}(\mathbf{p}_F, \mathbf{R}; z) = \alpha^*(-\mathbf{p}_F, \mathbf{R}; -z^*). \tag{11}$$

The quasiclassical Green's function is parameterized in terms of two scalar coherence functions, $\gamma(\mathbf{p}_F, \mathbf{R}; z)$ and $\tilde{\gamma}(\mathbf{p}_F, \mathbf{R}; z)$, as[59–65]

$$\hat{g} = -\frac{i\pi}{1 + \gamma\tilde{\gamma}} \begin{pmatrix} 1 - \gamma\tilde{\gamma} & 2\gamma \\ 2\tilde{\gamma} & -1 + \gamma\tilde{\gamma} \end{pmatrix}. \tag{12}$$

Note that with this parameterization, the Green's function is automatically normalized to $\hat{g}^2 = -\pi^2\hat{1}$. The coherence functions obey two Riccati equations:

$$\left(i\hbar\mathbf{v}_F \cdot \nabla + 2z + 2\frac{e}{c}\mathbf{v}_F \cdot \mathbf{A}\right)\gamma = -\tilde{\Delta}\gamma^2 - \Delta, \tag{13}$$

$$\left(i\hbar\mathbf{v}_F \cdot \nabla - 2z - 2\frac{e}{c}\mathbf{v}_F \cdot \mathbf{A}\right)\tilde{\gamma} = -\Delta\tilde{\gamma}^2 - \tilde{\Delta}, \tag{14}$$

where $\mathbf{A}$ is the vector potential. These first-order non-linear differential equations are solved by integration along straight (ballistic) quasiparticle trajectories. Quantum coherence is retained along these trajectories, but not between neighboring trajectories. A clean superconducting grain in vacuum is assumed by imposing the boundary condition of perfect specular reflection of quasiparticles along the edges of the system.

The superconducting order parameter is assumed to have pure $d$-wave symmetry

$$\Delta(\mathbf{p}_F, \mathbf{R}) = \Delta_d(\mathbf{R})\eta_d(\theta), \tag{15}$$

where $\theta$ is the angle between the Fermi momentum $\mathbf{p}_F$ and the crystal $\hat{\mathbf{a}}$-axis, and $\eta_d(\theta)$ is the $d$-wave basis function:

$$\eta_d(\theta) = \sqrt{2}\cos(2\theta), \tag{16}$$

fulfilling the normalization condition

$$\int \frac{d\theta}{2\pi} |\eta_d(\theta)|^2 = 1. \tag{17}$$

The order parameter amplitude satisfies the gap equation

$$\Delta_d(\mathbf{R}) = \lambda_d N_F k_B T \sum_{|\epsilon_n| \leq \Omega_c} \int \frac{d\theta}{2\pi} \eta_d^*(\theta) f(\mathbf{p}_F, \mathbf{R}; \epsilon_n), \tag{18}$$

where $\lambda_d$ is the pairing interaction, $N_F$ is the density of states at the Fermi level in the normal state, and $\Omega_c$ is a cutoff energy. The pairing interaction and the cutoff energy are eliminated in favor of the superconducting transition temperature $T_c$ (see for example ref. [66]) as

$$\frac{1}{\lambda_d N_F} = \ln \frac{T}{T_c} + \sum_{n \geq 0} \frac{1}{n + \frac{1}{2}}. \tag{19}$$

The above equations are solved self-consistently with respect to $\gamma$, $\tilde{\gamma}$, and $\Delta_d$. As an initial guess, we assume a homogenous superconductor with a small modulation of the phase. The coherence functions on the boundaries have to be updated in each iteration, taking into account the specular boundary condition. The starting guess is the local homogeneous solution. After several iterations, the information of the initial guess for the coherence functions is lost[67].

We choose an electromagnetic gauge where the vector potential has the form

$$\mathbf{A}_{ext}(\mathbf{R}) = \frac{1}{2} \mathbf{B}_{ext} \times \mathbf{R}. \tag{20}$$

The total vector potential $\mathbf{A}(\mathbf{R})$, that enters Eqs. (13) and (14), is given by $\mathbf{A}_{ext}(\mathbf{R})$ and the field $\mathbf{A}_{ind}(\mathbf{R})$ induced by the currents $\mathbf{j}(\mathbf{R})$ in the superconductor (Eq. (27) below):

$$\mathbf{A}(\mathbf{R}) = \mathbf{A}_{ext}(\mathbf{R}) + \mathbf{A}_{ind}(\mathbf{R}). \tag{21}$$

The vector potential $\mathbf{A}_{ind}(\mathbf{R})$ should be solved from Ampère's circuit law

$$\nabla \times \nabla \times \mathbf{A}_{ind}(\mathbf{R}) = \frac{4\pi}{c} \mathbf{j}(\mathbf{R}), \tag{22}$$

with appropriate boundary conditions for the induced field inside and outside the sample. To take the full electrodynamics into account, $A_{ind}(\mathbf{R})$ also needs to be computed self-consistently in each iteration. However, the strength of the electrodynamic back-coupling scales as $\kappa^{-2}$, where $\kappa \equiv \lambda_0/\xi_0$ is the dimensionless Ginzburg-Landau parameter. The electrodynamic back-coupling is therefore a very small effect for type II superconductors (typically $\kappa^{-1} \approx 10^{-2}$ for the cuprates). We have verified through fully self-consistent calculations that for grains with side lengths $D < \lambda$, as we limit ourselves to in this paper, it is always safe to neglect this back-coupling. For large system sizes, $D \gg \lambda$, back-coupling would ensure proper Meissner screening on the length scale $\lambda$ in the interior for $b < 1$ and the establishment of a proper Abrikosov vortex lattice with inter-vortex distances of order $\lambda$ for moderate fields $b > 1$, corresponding to field strengths of order $H_{c1}$. Since the spontaneous fields appearing below $T^*$ are located within a small distance of order $\xi_0 \ll \lambda$ from the boundary, the effect of back-coupling is small also in these cases. Only in very high fields, approaching $H_{c2}$, where inter-vortex distances become of order $\xi_0$ may we expect a serious effect on $T^*$, but this is beyond the scope of this paper.

The induced magnetic flux density is computed as

$$\mathbf{B}_{ind} = \nabla \times \mathbf{A}_{ind}. \tag{23}$$

We consider a layered superconductor with many weakly, for our purposes negligibly, coupled layers stacked in the c-axis direction. This ensures translational invariance in that direction. Therefore, we neglect the problem of the field distribution around the superconductor and focus on the field induced at the ab-plane where we have simply $\mathbf{B}_{ind} = B_{ind}\hat{\mathbf{z}}$.

**Gauge transformation**. Once the Green's function and the order parameter have been determined self-consistently, we can perform a gauge transformation in order to make the order parameter a real quantity and in the process extract the superfluid momentum $\mathbf{p}_s$. This can be illustrated by transforming the Riccati equation in Eq. (13). To begin with, the self-consistently obtained order parameter is complex, i.e.,

$$\Delta(\mathbf{p}_F, \mathbf{R}) = |\Delta_d(\mathbf{R})| \eta_d(\theta) e^{i\chi(\mathbf{R})}. \tag{24}$$

We make the ansatz

$$\gamma(\mathbf{p}_F, \mathbf{R}; z) = \gamma_0(\mathbf{p}_F, \mathbf{R}; z) e^{i\chi(\mathbf{R})}, \tag{25}$$

and put that into the Riccati equation. We obtain

$$[i\hbar \mathbf{v}_F \cdot \nabla + 2(z - \mathbf{v}_F \cdot \mathbf{p}_s)]\gamma_0 = -|\Delta_d|\eta_d(\gamma_0^2 + 1), \tag{26}$$

where $\mathbf{p}_s$ is defined in Eq. (1).

**Observables**. The current density is computed within the Matsubara technique through the formula

$$\mathbf{j}(\mathbf{R}) = 2\pi e N_F k_B T \sum_{\epsilon_n} \int \frac{d\theta}{2\pi} \mathbf{v}_F g(\mathbf{p}_F, \mathbf{R}; \epsilon_n). \tag{27}$$

In the results section, we shall show this current density in units of the depairing current

$$j_d \equiv 4\pi |e| k_B T_c N_F v_F. \tag{28}$$

The free-energy difference between the superconducting and the normal states is calculated with the Eilenberger free-energy functional[52]

$$\Omega_S(B, T) - \Omega_N(B, T) = \int d\mathbf{R} \left\{ \frac{\mathbf{B}_{ind}(\mathbf{R})^2}{8\pi} + |\Delta_d(\mathbf{R})|^2 N_F \ln \frac{T}{T_c} \right. $$
$$\left. + 2\pi N_F k_B T \sum_{\epsilon_n > 0} \left[ \frac{|\Delta_d(\mathbf{R})|^2}{\epsilon_n} + i\mathcal{I}(\mathbf{R}; \epsilon_n) \right] \right\}, \tag{29}$$

$$\mathcal{I}(\mathbf{R}, \epsilon_n) = \int \frac{d\theta}{2\pi} \left[ \tilde{\Delta}(\mathbf{p}_F, \mathbf{R}) \gamma(\mathbf{p}_F, \mathbf{R}; \epsilon_n) - \Delta(\mathbf{p}_F, \mathbf{R}) \tilde{\gamma}(\mathbf{p}_F, \mathbf{R}; \epsilon_n) \right]. \tag{30}$$

We have verified that this form of the free energy gives the same results as the Luttinger-Ward functional[26,55,64]. The entropy and specific heat capacity are obtained from the thermodynamic definitions

$$S = -\frac{\partial \Omega}{\partial T}, \tag{31}$$

$$C = T \frac{\partial S}{\partial T} = -T \frac{\partial^2 \Omega}{\partial T^2}. \tag{32}$$

**Data availability**. All relevant data are available from the authors.

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

## Acknowledgements

We thank the Swedish Research Council for financial support. It is a pleasure to thank Mikael Håkansson, Niclas Wennerdal, and Per Rudquist for valuable discussions.

## Author contributions

P.H. carried out the numerical calculations. P.H., A.B.V., M.F., and T.L. analyzed the results. P.H. and T.L. wrote the paper with contributions from A.B.V. and M.F.

## Additional information

**Competing interests:** The authors declare no competing interests.

