## [Peer Review File · Nature Communications]

Reviewers' comments:

Reviewer #1 (Remarks to the Author):

See attached report in PDF format.

Reviewer #2 (Remarks to the Author):

The authors calculate the superfluid momentum, the induced magnetic flux, and the superconducting order parameter in a nanoscale d-wave superconductor under a weak magnetic field, using quasiclassical theory. Although the order parameter is solved self-consistently, the induced vector potential is not, with the assumption of strong type-II behaviour. The authors examine in detail the appearance of "sinks", "sources", and "saddle points" in the supercurrent flow, which dominate over the paramagnetic Meissner effect near the surfaces, at low temperatures.

Although the work presented in the manuscript is interesting, it is merely a more detailed, follow-up study of what the same group has already published in Ref. 27 (Hakansson, Loefwander, and Fogelstroem, Nat. Phys. 11, 755 (2015)). In Ref. 27, using the same formulation in terms of quasiclassical theory, it has been shown that a second-order phase transition occurs at around $0.18 T_c$ (T_c is the superconducting transition temperature) to a state with a chain of vortex-antivortex pairs along the surface edges, spontaneously breaking time-reversal as well as translational symmetries. What the present manuscript shows is basically that a similar second-order phase transition occurs (at $0.18 T_c$) to a state with spontaneously broken translational symmetry with a chain of vortex-antivortex pairs along the surfaces, also in the presence of applied magnetic field, i.e., with broken time-reversal symmetry to begin with. Although there is asymmetry between vortices and antivortices due to the applied field and the authors use more fancy words such as sinks, sources, and saddle points, the main physics is the same -- a second-order phase transition to a phase where vortex-antivortex pairs appear along the surface edges, spontaneously breaking translational symmetry. The result similar to that in Ref. 27 is not surprising, considering the rather weak applied field (only up to one flux quantum over the entire system area).

Finally, as mentioned in the manuscript, the discussion on spontaneous appearance of supercurrents associated with the Andreev surface states in terms of quasiclassical theory itself is not new (Refs. 19-21).

The manuscript does not meet three of the four general criteria for publication in Nature Communications; novelty, extreme importance to scientists in the specific field, and interest to researchers in other related disciplines.

In summary, I do not recommend publication of the manuscript in Nature Communications. I believe that the manuscript is more suitable for more specialised journals such as Physical Review B.

Reviewer #3 (Remarks to the Author):

The manuscript reports a theoretical study of nonuniform d-wave superconducting (SC) states in a small grain. The authors recently predicted a phase transition with spontaneous breaking of time-reversal symmetry and continuous translational symmetry along the surfaces of the grain (Ref. 27 in the manuscript). This transition takes place below the transition temperature from the normal state to the trivial d-wave SC state preserving the two symmetries. The low-temperature SC phase is characterized by the formation of a chain of spontaneous current loops localized along the surfaces. In the present work, their theory is extended to the case with external magnetic fields. Numerical results of the current distribution in the grain, the temperature dependence of the total magnetic flux induced by an external magnetic field, etc. are presented in the manuscript. The authors discuss the observability of the phase transition on the basis of the detailed numerical results.

This is an interesting paper, which deserves publication in Nature Communications. Over all, the manuscript is well written. I recommend publication of the paper after the authors have addressed the

following point.

On p. 6, the authors state

"We shall neglect the problem of the field distribution around the superconductor..."

How this approximation can be justified? This approximation does not seriously affect the conclusion of the paper? The authors should give comments on this point.

Referee #1

Dear Editor,

I have considered the manuscript: “Broken Translational Symmetry at Edges of High-Temperature Superconductors”, by P. Holmvall et al. The authors report the theoretical prediction of a new superconducting phase in mesoscopic high-temperature superconductors with a pre-existing $d_{x^2 - y^2}$ order

parameter. The new phase is predicted to onset at a temperature of order $T^* \approx 0.18T_C$, where T_C is the superconducting transition temperature of the bulk superconductor.

The nature of the ordering, and thus the identification of an order parameter, is novel. The new phase is intrinsically inhomogeneous, exhibiting a combination of broken time-reversal symmetry (BTRS) and locally broken translation symmetry along the edge boundaries of the mesoscopic sample (a square geometry) in zero external magnetic field. Indeed the low-temperature phase in zero magnetic field (Fig. 1) is characterized by the configuration of 28 $n = 1/2$ vortices of $\vec{\mathfrak{P}}$ on the boundary, 15 $n = -1$ vortices organized along the [100] mirror planes and 1 $n = +1$ vortex in the center of the mesoscopic sample, which is net winding neutral. In addition, there are 4 corner sources whose winding number appears to be $n = +1/4$. It is a ground state that I would not have anticipated.

Nevertheless, I believe the analysis and results are correct and that the authors have found a distinctly new type of ordered phase in this class of unconventional superconductors. In contrast to previous theories for surface broken time-reversal symmetry there is no pairing channel other than the B_{1g} channel. The signature

of BTRS is a non-zero superflow field, $\vec{p}_s(\vec{R}) = \frac{1}{2} \nabla \chi - e/c \vec{A}$, where the vector potential, \vec{A} emerges with spatial phase variations, $\chi(\vec{R})$, of the local $d_{x^2 - y^2}$ order parameter in order to ensure local gauge invariance.

The authors identify the mechanism of the ordering as an instability originating from the large spectral weight of the zero-energy surface Andreev bound state that is characteristic of the $d_{x^2 - y^2}$ order parameter; i.e. when $N(0)$ at the edge is sufficiently large the free energy is lowered by splitting the zero-modes into

Doppler bands, defined by the emergence of \vec{p}_s , at finite energies above and below the Fermi level, with occupancy of the negative energy band lowering the energy. The reduction in free energy due to the Doppler splitting competes with the cost in kinetic energy associated with the edge currents such that the transition onsets when the zero-energy spectral weight becomes sufficiently large. What is less clear is why the ground state also breaks local translation invariance.

Our reply:

We outline our motivation for breaking of local translation invariance under the question labeled 4.2.12 below, since it is the same question.

Overall, the results reported of an intrinsically inhomogeneous ground state of d-wave high-temperature superconductors is both unanticipated and novel, and I believe will likely spur experimental investigations into mesoscopic unconventional superconductors. I recommend publication of the manuscript. The evolution of the phase with applied magnetic field provides an added dimension for exploring the nature of this phase transition. The senior authors are established experts on unconventional superconductivity, and in particular, inhomogeneous phases of this broad class of superconductors. Below are comments for the authors to consider for clarification of their results and presentation before publication. In the following the notation x.y.z refers to page x, paragraph y, line z.

Fig. (1): The winding numbers for the defects of \vec{p}_s shown in Fig. (1) appear in the caption and the abstract but are not explicitly defined in the text on page 2 where Fig. (1) is first discussed. I infer these to be winding numbers of a unit vector constructed from \vec{p}_s , rather than the winding number of the phase field $\chi(\vec{R})$. I would recommend the authors clarify the identification of winding numbers for the defects within the text.

Our reply:

The referee has correctly understood the figure. We have introduced the definition of the Poincaré index as Eq. (2). To avoid confusion, we have changed our nomenclature and now avoid the term “winding number”, since it is typically reserved for the winding of the order parameter phase.

2.1.16: On this same issue: (i) Can the authors clarify why the ground state in zero field appears to have a net winding of +1 with the assignments inferred from Fig. (1), plus $n = 1/4$ for the corner sources? (ii) Are the corner sources $n = 1/4$ defects? A net winding number of +1 would appear at odds with the result shown in Fig. (2) and Fig. (5a) in which the induced flux is zero for zero applied field. This would seem to suggest that the ground state should have a net winding number of zero for $B_{\text{ext}} = 0$.

Our reply:

We have improved our description of the topology of the vector field. In particular, the special points on the edges and the corners that can not be encircled by closed Jordan curves are now treated in a mathematically rigorous way through the generalized Poincaré-Hopf theorem [new Eq.(3)]. It states that the sum of the Poincaré indices of all internal critical points and a rule for counting the special points on the boundary leads to a sum rule related to the Euler characteristic. This beautiful sum rule is now illustrated in detail in the new Supplementary Information. After this change, we are confident that the confusion that the referee expressed here will be avoided.

Eq. 2: There is a mis-match in notation between Eq. (2) and the y-axis of Fig. 4 for the average current (or current density).

Our reply:

Indeed, there was a misprint in Fig.4 that we have corrected.

4.2.12: While the mechanism favoring $\vec{p}_s \neq 0$ is explained clearly earlier in the manuscript, there is no comparable explanation as to why the superfluid flow should also favor the breaking of local translational symmetry. The statement that the “... breaking of translational symmetry allows for a larger shift from zero

energy ...” is basically just re-phrasing the fact that the loop current configuration has lower energy. Thus, the reader is left with the question: “Why is it energetically favorable for \vec{p}_s to develop the loop current structure on the boundaries?”

Our reply:

From the present numerical calculations, we have to admit that we do not have a final answer to this question. We believe, however, that it is central to the problem that the midgap Andreev surface state wave function decays away from the surface on the coherence length scale ξ , actually it has decayed at $5*\xi$ from the surface. The $5*\xi$ can be converted to a temperature scale $T^*\sim 0.2T_c$. The quasiparticle current flow near the surface induced by the Doppler shift of the Andreev states, when p_s is finite in the symmetry broken state, will therefore flow near the surface as well. The question must be related to the back flowing supercurrents, i.e. the structure and origin of p_s itself. What sets the length scale for the supercurrents? Let us compare with earlier scenarios for time-reversal symmetry breaking, where local continuous translational invariance was enforced through assumptions.

The first one (Refs. 19-21) involves closing the equation system through Maxwell’s equations. In this case the back flowing screening currents flow by construction of the theory on a very long length scale, the penetration depth λ . The problem is that the induced p_s is then small, limited by ξ/λ . The resulting $T^*\sim(\xi/\lambda)T_c$ is small.

The second one (Ref. 22) is a slab geometry, with slab width D . This time, an effective Zeeman field due to confinement shift the bound states, but the shift is limited by the smallness of ξ/D . Still, $D<\lambda$, and $T^*\sim(\xi/D)T_c$ in this scenario is higher than in the first one.

In the present case, there is no new length scale entering the problem. If v_F*p_s (multiplying by the Fermi velocity to get an energy scale) in this case is simply limited by k_B*T_c , we may get supercurrents of the order of the depairing currents and T^* a substantial fraction of T_c .

One ingredient in getting loop currents is that the response function, the current response to p_s , should be highly non-local. This would translate the spatial dependence of the Andreev states (the tail of length $5*\xi$) to a response area of order $(5*\xi)^2$. If the response has off-diagonal components due to non-locality, a component perpendicular to the surface may be induced, and current loops of radius $5*\xi$ may form, as we see in the numerics presented in the manuscript. However, constructing and solving a complete response theory clarifying these points further is rather involved, and goes far beyond the scope of this manuscript.

Eq. 5: The authors might clarify the nature of the phase transition, at least in zero field, and the identification of an order parameter based on \vec{p}_s . The existence of a heat capacity jump is indicative of a second-order, mean-field transition, in which case one expects the order parameter to scale as, $p_s(T) \sim (1 - T/T^*)^\beta$ with exponent $\beta = 1/2$. However, in Fig. (4) the averaged current density appears to show a linear slope near T^* , as does the increase in $\bar{\Delta}_d$ shown in Fig. 5. It would be helpful to clarify, or at least state clearly, what the results are, and whether or not they agree with typical mean-field exponents.

Our reply:

We indeed find that the phase transition is a typical 2nd order phase transition, where the order parameter, i.e. p_s , grows below T^* with a $\beta=1/2$ exponent. However, nonlinearity sets in quickly for $T<T^*$ and the scaling is only obeyed in a very narrow temperature range. We have added a sentence at the end of the first paragraph, left column, on page 4 and an inset to Fig.6(d) to highlight this point.

Referee #2

The authors calculate the superfluid momentum, the induced magnetic flux, and the superconducting order parameter in a nanoscale d-wave superconductor under a weak magnetic field, using quasiclassical theory. Although the order parameter is solved self-consistently, the induced vector potential is not, with the assumption of strong type-II behaviour. The authors examine in detail the appearance of "sinks", "sources", and "saddle points" in the supercurrent flow, which dominate over the paramagnetic Meissner effect near the surfaces, at low temperatures.

Our reply:

We have actually computed the vector potential self-consistently for typical parameters of the model and found that it only gives negligible changes to the results. We have added text below Eq. (22) to clarify better that this back-coupling of the induced field is small and can be neglected.

Although the work presented in the manuscript is interesting, it is merely a more detailed, follow-up study of what the same group has already published in Ref. 27 (Hakansson, Loefwander, and Fogelstrom, Nat. Phys. 11, 755 (2015)). In Ref. 27, using the same formulation in terms of quasiclassical theory, it has been shown that a second-order phase transition occurs at around $0.18 T_c$ (T_c is the superconducting transition temperature) to a state with a chain of vortex-antivortex pairs along the surface edges, spontaneously breaking time-reversal as well as translational symmetries. What the present manuscript shows is basically that a similar second-order phase transition occurs (at $0.18 T_c$) to a state with spontaneously broken translational symmetry with a chain of vortex-antivortex pairs along the surfaces, also in the presence of applied magnetic field, i.e., with broken time-reversal symmetry to begin with. Although there is asymmetry between vortices and antivortices due to the applied field and the authors use more fancy words such as sinks, sources, and saddle points, the main physics is the same - a second-order phase transition to a phase where vortex-antivortex pairs appear along the surface edges, spontaneously breaking translational symmetry. The result similar to that in Ref. 27 is not surprising, considering the rather weak applied field (only up to one flux quantum over the entire system area).

Our reply:

There is no doubt that the Referee is an expert in the field, having correctly outlined many (but not all) of our results that we present in this manuscript. The results were judged interesting, but the conclusion mostly about novelty was judged less favorably due to seeming similarity with our previous work. We note that the other two Referees, on the other hand, have the opposite opinion.

Our opinion is that the identification of the order parameter vector field p_s leads to a whole new physical picture of what is going on here. We show that this order parameter has a non-trivial topology that is quite unique. Such textures have not been proposed before for singlet superconductors to the best of our knowledge. We therefore disagree with the referee that this is merely fancy words, but instead is a substantial step forward in the understanding of the physics, well beyond what was presented in Ref. 27. In fact, Referee 1 pointed out that the identification of the order parameter p_s and its structure is novel and not something that he/she had anticipated.

In addition, we have identified a number of experimental signatures of this phase transition, well beyond what was presented in Ref. 27. Indeed, Referee 1 stated that this distinctly new type of ordered phase and the results presented is likely to spur experimental investigations in these types of systems. Thereby the possible impact on the field was judged as big.

Finally, as mentioned in the manuscript, the discussion on spontaneous appearance of supercurrents associated with the Andreev surface states in terms of quasiclassical theory itself is not new (Refs. 19-21).

Our reply:

Although Refs. 19-21 involves supercurrents and associated Doppler shifts, they discuss a different phase transition, where the back-coupling with the electromagnetic field is important. Thereby the penetration depth enters and suppresses the transition temperature by a factor ξ/λ . In our work, this back-coupling is irrelevant. Instead, it is the topological edge defects that drive the transition. Actually, the phase transition in Refs. 19-21 can not occur, because it will be preceded by the phase transition that we discuss in the present manuscript.

The manuscript does not meet three of the four general criteria for publication in Nature Communications; novelty, extreme importance to scientists in the specific field, and interest to researchers in other related disciplines.

Our reply:

Our results are novel and of extreme importance to scientists in the specific field because

a) We have identified the order parameter: it is the superfluid momentum p_s in Eq.(1) that forms a beautiful planar vector field. It contains a number of sources, sinks, and saddle points. We have shown explicitly that the vector field satisfies a generalized Poincaré-Hopf theorem, thereby putting this aspect of the order parameter on solid mathematical grounds.

b) We have quantified in detail the thermodynamics in magnetic field and established a phase diagram in the Meissner state.

c) We have quantified a number of experimental consequences that can be directly tested in the lab, including non-monotonous temperature dependence of the induced magnetic fields at the edges measurable for instance with scanning SQUIDS, loss of the paramagnetic response below T^* that can be measured via the penetration depth, and jump in the heat capacity measurable with nanocalorimetry

Our motivation for that results are of interest to researchers in other related disciplines can be summarised as follows:

Today there is a broad interest into topological quantum materials, where the topology of the bulk guarantees edge/surface states at zero energy. However, it is well known that a large spectral weight of states exactly at the Fermi energy is energetically unfavourable. This competition between topological protection of a flat band of zero energy edge states and spontaneous symmetry breaking shifting these states to finite energy is a recently identified problem that has just started to be investigated (see e.g. also Ref. 40). In our manuscript we study this problem in d-wave superconductors with pair-breaking edges where there is a flat band of zero energy states. The study has however a broader experimental connection that can extend beyond d-wave, since there are current efforts in detection of (topological) bound states in other unconventional superconductors as well.

We would like to emphasize that in our manuscript we provide an illustrative example of the fragility of topologically protected flat bands of surface states, where the protection can be traced to certain symmetries. At the phase transition where those symmetries are spontaneously broken, the initial topological protection is lost. In addition, to the best of our knowledge, this is the first time that it has been shown that a singlet superconductor can support a phase with a new order parameter, the superfluid momentum p_s , having topological textures beyond the traditional Abrikosov vortices. Ironically, at the phase transition one topology is replaced by another type of topology!

This kind of topological vector field appears in many branches of physics, including general relativity, fluid dynamics, liquid crystals, and superfluid ^3He . We therefore believe that our results will be of interest for the wide audience of Nature Communications.

In summary, I do not recommend publication of the manuscript in Nature Communications. I believe that the manuscript is more suitable for more specialised journals such as Physical Review B.

Our reply:

With our motivation above that our manuscript meets the criteria of publication, and with the support of Referees #1 and #3, we hope the manuscript can indeed be published in Nature Communications.

Referee #3

The manuscript reports a theoretical study of nonuniform d-wave superconducting (SC) states in a small grain. The authors recently predicted a phase transition with spontaneous breaking of time-reversal symmetry and continuous translational symmetry along the surfaces of the grain (Ref. 27 in the manuscript). This transition takes place below the transition temperature from the normal state to the trivial d-wave SC state preserving the two symmetries. The low-temperature SC phase is characterized by the formation of a chain of spontaneous current loops localized along the surfaces. In the present work, their theory is extended to the case with external magnetic fields. Numerical results of the current distribution in the grain, the temperature dependence of the total magnetic flux induced by an external magnetic field, etc. are presented in the manuscript. The authors discuss the observability of the phase transition on the basis of the detailed numerical results.

This is an interesting paper, which deserves publication in Nature Communications. Over all, the manuscript is well written. I recommend publication of the paper after the authors have addressed the following point.

On p. 6, the authors state

"We shall neglect the problem of the field distribution around the superconductor..."

How this approximation can be justified? This approximation does not seriously affect the conclusion of the paper? The authors should give comments on this point.

Our reply:

We thank the Referee for the positive report, and the recommendation to publish in Nature Communications. For the question about the field distribution outside the superconductor, we clarify below Eq. (23) that we consider a layered superconductor with small interlayer coupling. It is then enough to consider one layer and then they are simply stacked in the z-direction creating translational invariance along the z-axis. As seen in the figures, the induced fields are confined to be within the grain. Therefore, we neglect any small stray fields for instance at the top and bottom faces of a real sample. We also added text below Eq. (22) to clarify that we have checked that the back-coupling of the induced field is a small effect within all parameter ranges we consider in this work.